# Metabolomics in Radiation Biodosimetry: Current Approaches and Advances

**DOI:** 10.3390/metabo10080328

**Published:** 2020-08-11

**Authors:** Merriline M. Satyamitra, David R. Cassatt, Brynn A. Hollingsworth, Paul W. Price, Carmen I. Rios, Lanyn P. Taliaferro, Thomas A. Winters, Andrea L. DiCarlo

**Affiliations:** 1Radiation and Nuclear Countermeasures Program (RNCP), Division of Allergy, Immunology and Transplantation (DAIT), and National Institute of Allergy and Infectious Diseases (NIAID), National Institutes of Health (NIH), 5601 Fishers Lane, Rockville, MD 20852, USA; cassattd@niaid.nih.gov (D.R.C.); brynn.hollingsworth@nih.gov (B.A.H.); carmen.rios@nih.gov (C.I.R.); lanyn.taliaferro@nih.gov (L.P.T.); twinters@nih.gov (T.A.W.); cohena@niaid.nih.gov (A.L.D.); 2Office of Regulatory Affairs, Division of Allergy, Immunology and Transplantation (DAIT), National Institute of Allergy and Infectious Diseases (NIAID), National Institutes of Health (NIH), 5601 Fishers Lane, Rockville, MD 20852, USA; paul.price@nih.gov

**Keywords:** radiation metabolomics, radiation biodosimetry, biomarker signature

## Abstract

Triage and medical intervention strategies for unanticipated exposure during a radiation incident benefit from the early, rapid and accurate assessment of dose level. Radiation exposure results in complex and persistent molecular and cellular responses that ultimately alter the levels of many biological markers, including the metabolomic phenotype. Metabolomics is an emerging field that promises the determination of radiation exposure by the qualitative and quantitative measurements of small molecules in a biological sample. This review highlights the current role of metabolomics in assessing radiation injury, as well as considerations for the diverse range of bioanalytical and sampling technologies that are being used to detect these changes. The authors also address the influence of the physiological status of an individual, the animal models studied, the technology and analysis employed in interrogating response to the radiation insult, and variables that factor into discovery and development of robust biomarker signatures. Furthermore, available databases for these studies have been reviewed, and existing regulatory guidance for metabolomics are discussed, with the ultimate goal of providing both context for this area of radiation research and the consideration of pathways for continued development.

## 1. Introduction

There is currently an urgent, unmet need for biodosimetry tests to detect radiation exposure levels, to be deployed in the event of a large-scale nuclear incident, and to monitor injury progression and recovery. To address this and other requirements, the Radiation and Nuclear Countermeasure Program (RNCP) within the National Institute of Allergy and Infectious Diseases (NIAID), National Institutes of Health (NIH) was initiated in 2004, with the mission to support research to develop medical countermeasures (MCMs) to diagnose (biodosimetry) and treat radiation injuries in the wake of a radiological/nuclear public health emergency. Currently, there are three medical countermeasures that are approved (as per FDA nomenclature, drugs are approved, biological products are licensed, and devices are cleared) by the U.S. Food and Drug Administration (FDA) and stockpiled by the government—filgrastim (Neupogen^®^, FDA approved March 2015; Amgen, Thousand Oaks, CA, USA) [1], pegfilgrastim (Neulasta^®^, FDA approved November 2015; Amgen, Thousand Oaks, CA, USA) [2], and sargramostim (Leukine^®^, FDA approved March 2018; Partner Therapeutics, Lexington, MA, USA) [3]. In contrast, no radiation biodosimetry tests have yet been cleared by the FDA for triage, dose assessment, or public health screening purposes.

The toolkit available to first responders and health professionals to respond to a radiological/nuclear incident will likely require multiple biodosimetry tests, such as (1) field-deployable methods to assess ≥2 Gy exposure in humans for triage, (2) laboratory-based, high throughput assays to determine definitive total body or partial body dose to the exposed individual [4,5], and (3) approaches to predict immediate and long-term consequences of radiation injury (Table 1). Apart from the classical cytogenetic assays (e.g., dicentric chromosome assay, micronucleus assay, and other DNA damage assessment methods), current technologies have expanded rapidly, to include proteomic, genomic, lipidomics, metabolomic, transcriptomic, and additional cytogenetic markers, such as gamma-H2AX foci [6].

The field of metabolomics has experienced tremendous growth in recent years. Riekeberg and Powers [7] noted that, ‘whereas genomics and proteomics identify what *might* happen, metabolomics identifies what *is* actually happening in the system”, and metabolomic data represent a more complete picture of the system’s response to insult (e.g., from infection, trauma, or radiation). Metabolomics (which includes lipidomics in this review) aims to detect small molecule (<1000 Da) biomarkers in biological samples, such as blood, urine, saliva, sebum, feces, tears, as well as organ tissue [8,9], which occur downstream of genomic, transcriptomic, and proteomic processes.

The field of radiation metabolomics attempts to correlate specific changes in metabolites to the level of radiation exposure. Some of the earliest studies relating metabolomics to radiation exposure were reported in the 1960s, with samples tested from irradiated rats [10,11] and humans [12,13]. With the emergence of innovative technologies, sophisticated and precision instruments, and data analysis tools that allow for a meaningful interpretation of the metabolomic signature, there has been a resurgence in radiation metabolomics in the past dozen years [14,15,16,17,18,19,20,21,22,23,24,25,26,27]. In 2011, Coy et al. reviewed the literature on radiation metabolomics, its application to radiation exposure and its potential as a lead biodosimetry test [16]. That review concluded that metabolomics had the potential to serve as an important means of accurately assessing radiation exposure. More recent reviews by Menon et al. [28], Roh [20], and Vincente et al. [29] have highlighted recent advances in radiation metabolomics as applied to biodosimetry. However, the metabolomics approach is a behemoth, with its own set of challenges ranging from specifics of sample preparation, technology and analysis employed, to the interpretation of output data and translation [30]. Despite technical advances in detection of these markers, the determination of exposure is hindered by heterogeneity of exposure due to variations in the radiation field, for example, total versus partial body exposures or radiological contaminations, which can skew biodosimetry signatures [31]. Hence, the authors have investigated the literature to ascertain best practices in the field of metabolomics, in an effort to provide a reference document for use by researchers, to further improve and accelerate the translation of this key technology. This review addresses research in radiation metabolomics, by exploring studies in a number of different disease and traumatic conditions, with emphasis on (1) rigor and reproducibility in discovery (untargeted) and applied (targeted) research, (2) the technology, analysis, and databases available for continuity of knowledge, and (3) the regulatory landscape.

## 2. Variability due to Irradiation Sources and Animal Model Selected

Irradiators: outcomes following a nuclear incident or terrorist nuclear attack are highly unpredictable, given that an improvised nuclear device or a ‘dirty bomb’ will comprise of high energy photon and neutron rays. To readily respond to a large-scale emergency of a radiological/nuclear nature, most radiation biodosimetry studies typically use self-shielded gamma irradiators (^137^Cs or ^60^Co), X-ray irradiators, or LINAC systems, in combination with small-animal adapted micro-computed tomography image-guided radiation therapy techniques, similar to those intended for diagnostic or therapeutic human use [32,33,34,35]. Most of the metabolomic studies referenced throughout this manuscript use X- or gamma rays as sources. Radiation exposures generated from γ-emitting sources originate from a sealed radionuclide source, ^137^Cs or ^60^Co, are monoenergetic in the order of megavolts (MV). X-ray irradiators such as the orthovoltage and megavoltage irradiators have higher energy capabilities that operate at 130–320 kV and 6–18 MV, respectively. The ability to penetrate tissues is dependent on the X-ray tube energy; greater X-ray energy will yield greater tissue penetration [35]. In the case of both sources, changes in metabolites such as citrulline, citric acid, creatine, taurine, carnitine, xanthine, creatinine, hypoxanthine, uric acid, and threonine were consistently observed (Table 2). Using a neutron source that simulates an exposure similar to that in Hiroshima at 1–1.5 km from the epicenter, investigators demonstrated a severe metabolic dysregulation, with perturbations in amino acid metabolism and fatty acid β-oxidation in irradiated mice [36]. Urinary metabolites were able to discriminate between neutron and X-rays on day 1, as well as day 7 post-irradiation. Therefore, much care should go into selection of the appropriate irradiation source, to ensure accuracy and sensitivity when planning advanced development of a radiation biodosimetry metabolomics panel.

Animal models of irradiation: radiation exposure victims will have variable radiation responses due to factors such as heterogenous exposures, resulting in random shielding that would likely spare a portion of tissue-regenerating stem and progenitor cells [37]. However, most of the radiation biodosimetry studies use total body irradiation (TBI) with homogenous exposures resulting in specific metabolomic panels (Table 2). Only a handful of radiation metabolomics studies utilize partial body irradiation (PBI) [38,39,40] with relevance to biodosimetry. Selection of the appropriate irradiation model for the intended use of the biodosimetry test is extremely important. For instance, a TBI model is appropriate for triage and definitive dose assay in the first few days following irradiation, although there is interest in understanding how partial body irradiation impacts the metabolomic signature during the early days post-exposure. However, for predictive biodosimetry for delayed effects of acute radiation exposure (DEARE), such as pulmonary, cardiac, or renal injuries, it is more appropriate to select a partially shielded model, to ensure the survival of the animals irradiated with high doses to 2–4 months post-exposure, to allow the manifestation of overt late effects [41].

The species used for these experiments are usually rodents (mouse and rats), and primates. Although most discovery work for a biodosimetry signature can be initiated in small rodents, their significant biological divergence from the human response presents considerable translational challenges. Larger mammal models, such as primates, canine or porcine species that have more synergy with human radiation response are more suited for bridging studies to demonstrate the effectiveness of a biodosimetry signature from animal data to relevant clinical metrics [42].

## 3. Bioanalysis: Rigor and Variability

As the scientific understanding of disease pathology and the mechanisms of action of therapeutics has advanced, especially in the realms of personalized medicine, understanding metabolomics data has become increasingly important, and these data can be used in several ways. The data can be used in a prospective manner, in the sense that certain patterns of small molecules appear to be able to predict later events, such as the evolution of lung pathology, cancer or diabetes after exposure to radiation. In a similar way, metabolomic signatures have been identified as useful diagnostic tools able to discern the presence of different disease states, based on the metabolome assessed in various sample types. The fluids commonly sampled for these analyses include whole blood, serum, plasma, urine, feces, saliva, and even breath and cerebrospinal fluid. Considerations for the detection of metabolic markers in these diverse biological samples will be discussed separately. There is also a myriad of confounders that can come into play when assessing the metabolic state of an irradiated human or animal model. Generally, these variables can be grouped into three categories: markers that are intrinsic and may be dependent on genetics, markers that are extrinsic and are dependent on environmental factors, and markers that are related to procedural testing. Careful consideration should be given to baseline genetics of the subject, circadian variations, age, sex, smoking status, frequency of exercise, diet, and more, when interpreting metabolomics data. The impact that each of these elements contribute to a well-designed study will be discussed below.

### 3.1. Intrinsic Factors

#### 3.1.1. Sex, Age and Ethnicity

Major determinants in the metabolic profile of an individual include age and sex [43]. Several studies have shown correlations between these variables and metabolites. For example, associations have been noted between levels of amino acids and acylcarnitines and age, sex and body mass index [44,45]. Not only do age- and sex-based differences in the metabolome exist, they also form the basis of differential responses to other stimuli, such as radiation exposure or traumatic injury. On investigating samples from total body irradiated (TBI) humans, Laiakis et al. noted sex-dependent differences in the biomarker signature, and cautioned that separate metabolomic panels for males and females be deployed as radiation biodosimetry tests [22]. Lusczek et al. found that not only were differences in the metabolomic profile attributable to aging, adding trauma to the equation, which produced dramatic responses, still allowed for the differentiation of samples into different ages and sexes [46]. Age has also been noted as a factor that should be considered in evaluation of radiation-induced metabolomic signatures [47]. Although there are notable changes in the metabolomic signature as we age, the most dramatic changes have been noted in infants within the first year [48]. In another study [49], a metabolic profile generated using urine and plasma samples permitted the highly accurate prediction of sex and age of the test subjects. Fecal samples have also been used to detect signatures indicative of sex in some disease states [50]. The ethnicity of the subject can also have an impact on metabolic signatures. For example, differences in the profiles of individual smokers were noted between African Americans and Caucasians [51].

It is, presumably, easier to account for confounding factors when planning a pre-clinical study. For example, selection of an inbred strain, with all animals born within a few days of each other, can cut down considerably on genetic factors that might be in play. Sex-correlated differences in animal models have also been noted [52], with sex widely believed to be the most important variable in these studies.

#### 3.1.2. Disease

In studies that involve both predictive and retrospective/diagnostic metabolomics, the presence of certain metabolite signatures has been linked to both the increased risk of, and the likely presence of cancer in the body [53]. It is likely that the metabolic pathways that are active in cancer cells, which allow the tumor to stay perfused and continue to grow, involve the production of biomarkers that can then be used to monitor tumor status and efficacy of therapies. These studies have indicated a correlation between metabolite profiles and disease in cancers, such as breast [54,55,56], endometrial [57], gastrointestinal (GI) [58], colorectal [59], head and neck [60], pancreatic [61,62], and bladder [63].

Diabetes, perhaps because it is so tightly linked to diet and obesity, is another disease state that has been carefully studied by metabolomics researchers. This research has included both prospective work, which suggests that it may be possible to predict the risk of developing type-2 diabetes [64], and the rate of progression [65], as well as the detection of correlative biomarkers, including fatty-, amino- and bile-acids, identified in patients with the disease. Levels of hemoglobin A1c specifically are linked to the pathophysiology of the disease [66]. There is also evidence that several chronic diseases and afflictions as wide and varied as autism [67], dengue infection [68], and chronic obstructive pulmonary disease [69] have distinct signatures, which display a sex-specific array of metabolomic phenotypes. It is likely that these preexisting metabolomic alterations can impact the radiation biodosimetry-specific panels that are usually generated in healthy animal models.

### 3.2. Extrinsic Factors

Smoking- as mentioned above, environmental factors can have a considerable impact on the metabolic signature of an individual. For example, smoking is known to alter DNA methylation, with the finding that there are unique epigenetic changes that are attributable to the habit [70,71]. The examination of exhaled breath profiles in smokers versus non-smokers further reinforces these findings, with results that show clustering of spectra attributable to smoking that are easily detectable [72]. In terms of the impact of smoking on the evolution of responses to irradiation, although there are few studies that have established a link between smoking and radiation-induced alterations in the metabolome, it is reasonable to expect that a person’s smoking status, to the extent that it alters baseline metabolites, should be considered as a possible confounder in radiation studies.

Obesity- Another environmental/lifestyle factor that is known to play a major role in someone’s metabolic signature is diet and obesity. In fact, findings in this area suggest that not only are the metabolic profiles different between normal weight and obese individuals [73,74], but there are certain metabolic profiles that are predictive of an individual’s propensity to become obese [75]. Obesity is then further tied to the development of type 2 diabetes, with biomarker assessment allowing for a better understanding of the mechanisms that might be involved in the development of the disease in overweight individuals [66]. In terms of the impact of diabetes and obesity in the establishment of radiation signatures, one study in mice demonstrated that in mice fed a high-fat diet and then induced to develop diabetes, there were definitive changes in the effect of whole-body, low dose radiation exposure on resulting diabetes-induced renal injury, as well as kidney lipid profiles [76]. Furthermore, in atomic bomb survivors, radiation exposure changes in the metabolic profile in the context of inflammation related to obesity were also noted [77]. For these reasons, there is excitement in the field of metabolomics that these markers may be able to elucidate the mechanisms involved in development of obesity, for which interventions might be targeted.

Exercise- Closely tied with differences in metabolites between overweight and lean individuals are the differences noted in sedentary animal models and those that are exercise trained [78]. Adding to this complexity is the need to consider if an individual has been active prior to providing a sample for analysis. For example, a study that looked at changes after exercise [79] found that activity routinely altered levels of metabolites that were linked to energy metabolism. Other studies have found large (>2-fold) changes in lipid metabolites within a few hours of exercise, which returned to normal levels within 24 h [80]. It would also appear that the intensity of exercise can be a factor in resulting metabolomic changes, with high-load and low-load resistance activities yielding different profiles [81]. In a rat model, changes in exercise-induced metabolites were not limited to the serum metabolome, as muscle, liver and heart tissues were also found to experience metabolic changes that were different from each other [78]. Furthermore, in a similar rat model, sampling from sedentary and active animals showed different results depending on how long the animal exercised [82]. There is also evidence that there is a change in metabolites in breast cancer patients that are subject to an exercise routine while undergoing radiation therapy [83]. Those women that were randomized to a 12-week resistance exercise group in a clinical trial were found to have lowered levels of certain kynurenic acid pathway urinary metabolites known to promote cancer progression, which were found to be elevated in irradiated patients in the control group. These findings suggest that, not only can exercise modify the radiation metabolome, it can also be beneficial in the course of cancer treatment.

Stress and mental illness- Another aspect of metabolic change that has not been as well studied as other variables is that of stress. There is evidence that post-traumatic stress disorder (PTSD) leads to the levels of circulating metabolites that suggest mitochondrial dysfunction, in both humans and animals [84]. Psychological stress has also been found to generate metabolic profiles that are significantly different from those seen for unstressed individuals, and that are implicated in cardiovascular conditions like heart disease [85]. Understanding the impact of stress on metabolism is extremely important in a radiation scenario, since there will definitely be high levels of psychological strain in individuals who survive a radiological or nuclear incident. Another related area of research that has determined correlations between metabolomic signatures and mental health outcomes is in depression. A recent meta-analysis [86] found strong associations between lipid metabolites and the prevalence of a depressive state. In fact, there are studies that suggest that the presence of certain metabolic markers can be used to predict both the propensity to develop depression [87], as well as the ability of patients taking anti-depression medications to demonstrate recovery [88]. Another group found that there were significant differences in the parts of the metabolomic profile associated with energy metabolism in the brain that could be linked, along with protein changes, to major depressive disorders [89].

Nutrition- It is not surprising that the food that someone is eating can have a major impact on a metabolomic profile; in fact, urine metabolites have been shown to correlate with obesity in rats fed a high fat diet [90]. The popular ketogenic diet has also been a recent area of research, with the finding that biochemistry and epigenetic changes have been observed [91], including those that appear to improve with the ketogenic plan. In that study [92], which used a mouse model with a breast-cancer xenograft, the authors found that, when a ketogenic feeding was combined with chemotherapy, animals on the special diet had a different metabolomic signature than tumor-bearing mice on a control diet, and those signatures more closely resembled that of the healthy mice (without a tumor). Another popular food plan, the “Mediterranean Diet”, has also been studied for its impact on both the metabolomic signature and microbiome of humans [93]. In a review of clinical trial information, beneficial effects of the diet were seen, and were believed to be mediated by changes in the gut microbiome in response to the altered nutrition. In a similar way, high-sugar diets, using a comparison of high and low fructose foods, yielded a difference in metabolomic profiles in the lean and obese women who were studied [94]. Even the vegan diet has been shown to lead to dramatic changes in the plasma metabolome [95]. Similarly, dietary studies done in nonhuman primates (NHPs) demonstrate changes in the metabolomics in monkeys fed different diets; however, there are also distinct patterns in the metabolomics between species, which appear to be consistent with different enzymes involved in metabolism [96]. Surprisingly, even something as innocuous as coffee consumption can lead to identifiable alterations in the circulating metabolites in the serum [97]. In that study, done in a rat strain that has a propensity toward diet-induced obesity, researchers found that the levels of certain fatty acids were modified, with the coffee consumption presumed to play a role by altering the microbiome of the gut. The impact of certain dietary changes in the radiation response of both circulating blood [98] and irradiated skin [99] has also been noted. For example, in ex vivo irradiated blood samples from healthy donors who only modified their diet through increased consumption of tomato juice (known to contain free radical scavengers), there were changes noted in the levels of plasma reactive metabolite compounds and concentrations of lycopene and β-carotene. In a study of rats fed a high-fat diet, the animals had higher resistance to radiation-induced skin injury than animals on a control diet, a finding that was linked the abundance of palmitic acid in the tissues of the high-fat diet animals.

### 3.3. Procedural Testing

Circadian rhythm—As with nearly all biological functions, circadian rhythm can influence the outcomes noted in studies of metabolomics [100,101]. These differences have been noted for many different types of samples, including urine, serum, saliva and breath, with findings of circadian variations in almost every metabolic pathway [102]. Sometimes referred to as “CircadiOmics” [103], there are demonstrated changes in a number of variables, including genomics, transcriptomics, proteomics and metabolomics. Because these differences are known, it is critical that they be considered during the design stage of any study, so as to limit their impact on data that are obtained. These circadian changes can be further modified based on nutrition [104]. It has long been known that the body’s response to radiation also falls into predictable patterns based on the time of day of the exposure [105,106]. In addition, there are day-to-day variations seen in the samples collected from serum and urine, with some types of metabolites showing greater variability than others [107]. For example, lipoproteins appear to be more stable longitudinally in a study than other metabolites. In addition, procedures that are used to collect (e.g., what part of the urine stream is collected) and process (e.g., timing, storage, and centrifugation) samples could have a subtle effect on the resulting data.

Predictive metabolomics—At one extreme of the predictive use of metabolomics is research that looks at metabolomic profiles in children (e.g.,—children who were breast-fed versus those that were not) [108], to determine if there is a signature present at a young age that could be used to predict some later aspect of health and development. The reports of prospective use of these kinds are markers are varied in scope, but a small sampling of the literature finds studies that indicate that there are also signatures that contain predictive markers of the progression of amyotrophic lateral sclerosis [109], bone mass in menopausal women [110] and a proclivity towards the development of major depression [87]. Of perhaps great relevance given the coronavirus pandemic is the finding that predictions about susceptibility to acute respiratory distress syndrome (ARDS) might also be made using metabolomic markers [111]. That research found that a biomarker panel consisting of proline, lysine/arginine, taurine, threonine and glutamate that were “characteristic of ARDS sub-stages. Similarly, there appear to be metabolomic markers (part of selected multi-metabolite panels) in the tissues of irradiated NHPs that can accurately (>90%) predict animal survival and the development of delayed radiation injuries in organs such as the lung, heart, kidney and brain [112].

## 4. Types of Metabolomic Samples

With radiation injury being both multi-organ and incredibly diverse across individuals, research into a variety of sample types is paramount to understand the progression of various sub-syndromes and organ injury. Additionally, with the large number of individuals seeking care in a mass casualty scenario, and the fragile state of the most affected, minimally invasive sample collections, like urine, saliva, feces, or a single drop of blood is preferable over a large blood draw or other more invasive sample collection procedures. Radiation-induced metabolomic changes in blood, urine, saliva, and feces from mice, rats, NHPs, and humans have been explored, and are reviewed below. Protocols routinely used for the preparation of samples including for liquid chromatography mass spectrometry (LC-MS) metabolomic analysis were published by Laiakis et al. [21], as was a method which allows for the possibility of rapid preparation protocols for blood [113]. Vincente et al. [29] recently published a detailed tabulation of different metabolite trends from various biosamples across multiple species, radiation doses, and time-points.

### 4.1. Blood

Blood is one of the most commonly used biofluids for any sort of testing, and is an obvious choice to investigate for metabolites that may assist in dose reconstruction, injury diagnosis, and prognosis. Indeed, blood metabolomic markers may be able to distinguish the type of radiation one is exposed to, as was found when mice were exposed to X-ray vs. neutron radiation [36]; or determine specific organ injury, as Jones et al. found when studying small intestine tissue metabolites compared to plasma metabolites of mice exposed to TBI [114]. Importantly, some metabolite signatures, including increases in polyunsaturated fatty acid-containing lipids, have been consistently found across mice [115,116], NHPs [26,117], and humans [118], over a range of radiation doses. Not only is blood a useful biosample to assess acute radiation response, but can also be used to predict delayed effects of acute radiation exposure (DEARE). Boerma and colleagues have reported long term changes in plasma and tissue metabolites in irradiated rats and humans, indicative of radiation-induced heart disease [119], presenting primarily as dyslipidemia. While commonly used and reliable, blood draws can be invasive and dangerous in already fragile patients with acute exposure to radiation.

### 4.2. Urine

Unlike blood, urine is an easily and painlessly collected biofluid for most animal models and humans. For this reason, urinalysis has been conducted for many years, with metabolite and other changes in urine used to detect any number of conditions, ranging from diabetes to pregnancy [120,121]. Changes in urine metabolites in response to irradiation have been known for decades, with rodent studies first conducted in the 1980s [122]. A wide variety of metabolic pathways can be impacted by radiation, as seen in the urine, with changes in biomarkers reflecting damage or changes to the microbiome, urea cycle, as well as amino acid, protein, fatty acid, and hormone metabolism across rodents [123,124], NHPs [125,126,127], and humans [22]. Curiously, Laiakis et al. found that metabolic markers in mouse urine could distinguish between lipopolysaccharide (LPS)-induced inflammation and radiation-induced inflammation [128], even different types of radiation were distinguishable in these samples [24,36]. Some urine metabolomic biomarkers like creatinine could even be used to differentiate sub-syndromes or specific organ injury, for example, creatinine levels can indicate kidney damage at high dose rates in NHPs [129]. Radiation-induced changes in urinary metabolomic biomarkers have also been found to be consistent across species. In a series of papers published by the Idle group, urinary metabolomics biomarkers were explored using mass spectrometry methods in mice [14,15], rats [130,131], and NHPs [18]. Biomarkers such as taurine, 2′-deoxyxanthosine, 2′-deoxyuridine, thymidine and *N*-hex-anoylglycine were identified, and were found to have similar responses to radiation across species. Various carnitines including acetylcarnitine were also found to be elevated following irradiation, in both NHPs [132] and humans undergoing fractionated TBI [22,133]. Xanthines were also elevated in the urine following irradiation, indicating DNA damage in both NHPs [126] and humans receiving TBI [22]. The ease of urine collection and consistency in findings across species make it a useful and promising biofluid to explore further for acute radiation exposure dose reconstruction, injury diagnosis, and prognosis.

### 4.3. Saliva

Saliva is also an important and promising biofluid to analyze for radiation-induced metabolomic changes. The salivary glands are a radiosensitive tissue [134], and are thus susceptible to ingested radioactive iodine, since the striated ductal cells express sodium iodide symporter [135]. Additionally, saliva metabolites have long been known to reflect plasma metabolites [136,137,138,139]. While the metabolomic responses to radiation have not been studied as extensively in saliva as in other biofluids, levels of metabolites such as amino acids were found to be responsive to TBI in the saliva of mice [23] and NHPs, where the response in NHPs lasted up to 60 days post-irradiation, though the investigators did not ascribe any ‘predictive’ role to this observation [140]. No studies have been conducted on the saliva of humans receiving fractionated TBI, and saliva is often difficult to collect post-radiation from patients who have received head and neck irradiation, due to xerostomia being a common side effect of radiotherapy [141]. Although xerostomia and sialadenitis are also common side effects of radioactive iodine-131 therapy for thyroid cancer [142], Wolfram et al. found that prostaglandin 8-epi-PGF2 alpha showed a significant ^131^I dose-dependent transient increase, though this increase was exacerbated by smoking [143]. Further investigation into the saliva metabolome pre- and post-acute radiation exposure could yield interesting biomarkers of dose exposure and injury.

### 4.4. Feces

Fecal metabolomic changes may be of interest to determine the extent of radiation-induced damage to the GI tract, a particularly susceptible system due to its high cell turnover rate and vital crypt stem cells for maintaining intestinal lining integrity and function [144]. Not only is it appropriate to use metabolomic biomarkers to determine intestinal lining damage, it is also possible to determine disturbance to the gut microbiome, which is essential for GI health. Indeed, Goudarzi et al. reported significant changes in not only bile acids such as taurocholic acid and 12-ketodeoxycholic acid, but also in microbial-derived products, such as pipecolic, glutaconic and homogentisic acids, and urobilinogen in feces collected from mice pre- or post-irradiation [145]. While nuclear incidents are thankfully rare, the assessment of feces collected from patients receiving TBI or even thoracic, abdominal, or pelvic radiation may be helpful in exploring these metabolomic changes. As in mice, Chai et al. also found changes in concentrations of bile acids in fecal samples, following pelvic irradiation in cervical cancer patients. Changes in the levels of other metabolites were also identified, including increases in α-ketobutyrate, valine, uracil, tyrosine, trimethylamine N-oxide, phenylalanine, lysine, isoleucine, glutamine, creatinine, creatine, aminohippurate, and alanine, as well as decreases in α-glucose, n-butyrate, methylamine, and ethanol [146]. Feces of patients who developed acute intestinal symptoms were also compared with feces of patients who did not develop these symptoms, with trimethylamine, n-butyrate, fumarate and acetate decreased and valine, trimethylamine N-oxide, taurine, phenylalanine, lactate, isoleucine and creatinine increased in those with GI symptoms [146]. The latter type of study may be useful in identifying biomarkers for triaging exposed individuals following a nuclear incident.

### 4.5. Other Types of Biofluid (Sebum, Sputum, Breath, Tears)

A few sample types that may be of interest have not yet been fully explored in the metabolomic response to radiation exposures. While rat sebum was found to be unaltered at 1or 24 h post-3 Gy of gamma-irradiation [147], sputum, and exhaled breath condensate (EBC), could be explored further particularly outside of the lung cancer space. Sputum may be of specific interest, given the delayed effects to the lung that can be generated following radiation exposure. Methods, approaches, and lessons-learned may also be gleaned and applied to the radiation biology field, from glaucoma research looking at metabolomic profiles in tears [148] to lung cancer work utilizing EBC and sputum changes across time [149]. For example, Ahmed et al. found elevated levels of propionate, ethanol, acetate, and acetone in the EBC of patients with lung cancer, compared with patients who had benign respiratory conditions, although curiously, propionate and acetate were found to be decreased in the sputum of lung cancer patients, along with glucose, N-acetyl sugars, glycoprotein, lysine, and formate [150]. Phillips et al. demonstrated an increase in 58 volatile organic compounds (VOC), consisting mainly of derivatives of alkanes, alkenes and benzene in irradiated Gottingen minipigs [151]. In another report, breath tests in 31 patients exposed to daily fractionated doses of 1.8–4.0 Gy or high doses of stereotactic body radiotherapy (7–12 Gy) identified 50 VOCs, of which 15 VOCs were persistently expressed for 5 days post-exposure, and could identify doses of 1.8 Gy and higher [152]. Fedrigo et al. also explored VOCs in radiotherapy patients before and after radiotherapy [153].

Radiation injuries can rapidly and constantly change over time, dependent upon the level and quality of the radiation exposure, treatment methods and other variables [19,25]. Thus, it is extremely important to collect samples longitudinally to track these functional metabolomic changes, in order to determine progression of injury and timepoints with greatest diagnostic and prognostic accuracy [123]. Temporal changes in metabolomic biomarkers can also vary depending on sex [22,118,126,154], as discussed earlier. Additionally, temporal metabolomic changes and diagnostic/prognostic accuracy of metabolomic markers may vary depending on the sample type [127,140], which is also a factor to consider when designing biomarker discovery and validation studies.

## 5. Metabolomic Biodosimetry Technologies

Early detection of radiation exposure allows for streamlined triage and subsequent clinical evaluation for planning appropriate medical intervention. The optimum detection technology would be rapid, accurate and easy to use with minimal training. Evaluation of biological manifestation (clinical symptoms and hematopoietic effects) of radiation exposure is the current method used in diagnosing acute radiation exposure. With the appropriate assessments, biodosimetric devices can assist in reconstructing the ionizing radiation dose received by an individual, using physiological, chemical, or biological markers that provide quantitative and/or qualitative outputs. However, immune system variability across the human population and the need for multiple assessments creates major challenges in identifying individuals who may require higher levels of medical intervention [42]. The abundance of metabolites, such as those generated by nucleotide deamination, lipid peroxidation, and activation of radical scavenging biomarkers, are unbiased, and their abundance is dependent on radiation exposure level rather than genetic differences [155].

The metabolomic profile of an organism is complex and may require different techniques to identify the variety of metabolites present in a sample [156]. For example, samples of *Saccharomyces cerevisiae* contain approximately 600 metabolites [157], the plant kingdom has an estimated 200,000 primary and secondary metabolites [158], while the human metabolomic profile easily exceeds these numbers. In an effort to study the complex metabolome, two technologies—nuclear magnetic resonance (NMR) and mass spectrometry (MS)—have been used extensively in the field. NMR has lower sensitivity, but has superb reproducibility, and is able to quantify more abundant compounds present in biological fluids, cell extracts, and tissues. It can also be used to identify structures of unknown compounds. One-dimensional (1D) ^1^H NMR is the most widely used NMR approach in metabolomics. Peak identification can be facilitated by the use of rNMR, a software package that automates metabolite identification [159].

NMR has been used to study the urine metabolite profile in radiation survivors of the Chernobyl Nuclear Reactor accident. Changes in the NMR peaks could be used to quantitatively compare changes in metabolites of survivors compared to normal controls. Variations in the metabolite concentrations noted in survivors were considerable, and some distortions were seen as a result of water suppression and the use of deuterium. In addition, some metabolites such as methylene were pH sensitive, leading to misassignments of resonances [160]. Overall, NMR offers some advantages that include non-destructive sample analysis, high reproducibility, and the ability to perform accurate quantification [154,161]. On the other hand, NMR offers low sensitivity, requiring a large sample volume, as well as the identification of high abundance metabolites. In fact, multiple strategies may be needed to characterize different metabolites. Moreover, highly complex spectra require meticulous peak deconvolution, making NMR time consuming [159]. The use of NMR has fallen out of favor in the field of metabolomics, as marked by the increase in publications that use MS-based approaches. However, NMR proponents believe that NMR is underappreciated, and both technologies should be used synergistically [159].

MS has been used in the field of radiation metabolomics since the 1970s to study metabolites in human urine and tissue extracts [161,162]. MS offers more sensitivity and can allow for higher throughput [159]. Highly quantitative, accurate and well-resolved metabolite profiles can be obtained depending on the type of MS technology used [163]. A variety of MS platforms exist, including: (1) liquid chromatography-tandem MS (LC-MS/MS), which offers excellent specificity and sensitivity providing accurate quantification [164]; (2) Fourier-transform ion cyclotron resonance (FTICR), which provides superior mass accuracy and resolution; (3) and desorption electrospray ionization (DESI) that can help spatially resolve metabolite profiles [165]. Furthermore, background noise can be minimized and metabolomic coverage can be increased by the use of complementary separation techniques, such as such as ultra-performance liquid chromatography (UPLC), gas chromatography (GC), and capillary electrophoresis (CE) [166].

It is important to understand the strengths and limitations of each method, in order to optimize the identification of the appropriate untargeted metabolites. In a study comparing CE–MS and LC–MS metabolite profiles from urine, approximately 500 metabolites were detected by CE-TOF–MS, while UPLC–MS detected only 300 metabolites, and the metabolite sets generated by each technology were different. Although the metabolite sets differed, they appeared complementary. CE–TOF-MS was better able to detect highly-polar metabolites, with a mass to charge value (m/z) ranging from 50–150, and UPLC-MS was better at finding metabolites with m/z above 150 [167]. While MS provides a reliable platform for the discovery of metabolomic biomarkers in multiple species (e.g., mice, rats, NHPs, and humans), different techniques have varying sensitivities, specificities, and detection limits. Below are some examples of MS technologies that have been used to discover and identify metabolites. None of them are superior to any of the others, but each offers different strengths and detection limits.

### 5.1. Gas Chromatography Mass Spectrometry (GC-MS)

GC-MS is considered by many to be the gold standard in metabolomics, but it is biased against non-volatile, high molecular weight metabolites. To capture these volatile metabolites, samples must be chemically processed at room or higher temperatures to allow for thermal stability and volatility. Sample hygrometry can also affect the stability of a sample; therefore, care must be taken to avoid generation of chemical breakdown components. During analysis, small aliquots (~1 uL) are injected into GC columns of differing polarity, which facilitate chromatographic resolution at pmol or nmol concentrations [168]. To aid in peak separation during spectral acquisition, time-of-flight (TOF) and quadrupole (Q) mass analyzers are often applied in GC-MS metabolomics. Both options provide faster and more high-throughput analysis. The GC-MS platform is relatively easy to use and considered cost-effective. It also provides great metabolite resolution, and the identification of peaks can be done quickly by using readily available mass spectral databases.

In a recent study, changes in urinary and serum metabolites were examined in four NHPs that had been exposed to 4 Gy γ-ray TBI (LD_50/60_ ~6.6 Gy). GC-TOF-MS was utilized to determine the kinetics of the metabolomic profile of NHP biofluids from pre-exposure to 60 days post-irradiation. Changes were found in the amino acid purine, serotonin, and lipid metabolism. These alterations could reflect changes in diet and/or the microbiome in response to the radiation insult. In addition, significant perturbations were found in the tricarboxylic acid (TCA) cycle at D3, a potential sign of mitochondrial dysfunction, with a return to baseline between D5 to D7 [154].

### 5.2. Liquid Chromatography–Mass Spectrometry (LC-MS)

Unlike GC-MS, LC-MS does not require sample volatility, and can be done at lower temperatures, simplifying sample analysis. LC-MS has proven to be valuable due to its metabolite selectivity and sensitivity; however, chromatography is time-consuming and may require several pre-treatment and post-treatment steps, which complicates sample processing and increases the amount of whole blood (100–200 μL) needed. One major disadvantage is that polar metabolites are poorly retained by high-performance liquid chromatography (HPLC) columns and elute with the solvent peak, limiting the metabolite data. Working with polar metabolites can be challenging and requires exploratory studies on solid and liquid phase combinations. If used in tandem with electrospray ionization (ESI), sample derivatization may be necessary. ESI only recognizes metabolites with positive or negative ionization, therefore, to identify a wide range of metabolites, analysis should be performed in both modes (positive and negative) [169]. Other peak separation techniques used in LC-MS metabolomic studies are HPLC, ultra-performance liquid chromatography quadrupole time-of-flight (UPLC-QTOF), and high-throughput, fabric phase sorptive extraction (FPSE).

The coupling of HPLC for physical separation and mass spectrometry (MS) for mass analysis is a long-standing technique in metabolomic studies. The method has proven to be valuable due to its metabolite selectivity and sensitivity, although the chromatography step is time-consuming and is limited in the number of assessments that can be done. By adding a second mass spectrometry detector (tandem mass spectra; MS/MS), selected molecules can be further analyzed to detect impurities, measure concentration levels of both identified and unidentified compounds, and provide information on chemical structures. Method and software development for interpreting tandem mass spectra has advanced, such that qualitative and quantitative information on RNA modifications can be determined, often at the level of sequence specificity.

High-throughput, FPSE minimizes processing and stabilization of whole blood without serum separation. FPSE uses a matrix coated with sol–gel poly (caprolactone-*b*-dimethylsiloxane-*b*-caprolactone) that binds polar and nonpolar metabolites. Taraboletti and colleagues have shown that the FPSE preparation technique, when combined with LC-MS, can differentiate radiation exposure markers (i.e., taurine, carnitine, arachidonic acid, α-linolenic acid, and oleic acid), measurable after 24 h post-8 Gy irradiation. The team suggests that this approach of biomarker screening, stabilization of biofluids between collection, and sample analysis could be an effective tool to triage individuals exposed to radiation during a radiological or nuclear incident [113].

Tissue damage resulting from radiation exposure produces small molecule signatures that can point to the severity of the acute radiation syndrome (ARS). Identification of radiation-induced changes in NHP metabolites can allow comparison to the effect on humans and can serve as a tool for biodosimetry. A study using global metabolomics aimed to measure biofluid metabolites impacted by ionizing radiation over time, by measuring metabolite levels over 60 days in urine and serum using an LC-MS approach in irradiated NHPs (4 Gy, TBI). A panel of metabolites was identified from previous studies that demonstrated significant alterations at 7 days post-4 Gy TBI. This finding led to the development of a rapid LC-MS/MS with multiple reaction monitoring (MRM) assay, for simultaneous absolute quantification in biofluids. Using this technique, eight compounds involved in protein metabolism, fatty acid β- oxidation, DNA base deamination, and general energy metabolism were identified, while a combination of three serum compounds and two urine compounds were shown to have excellent sensitivity and specificity for classifying exposures at 7 days post-4 Gy. These data highlight the dynamic nature of metabolites after radiation exposure, the value of MS-based techniques, and the need for timely biodosimetry [127].

In another study, untargeted metabolomic profiling was conducted. A combination GC-MS and targeted amino acid profiling based on LC-MS/MS were used to investigate early responses of metabolites from rat urine within 48 h post-total body exposure. Male Wistar rats were exposed to 2, 4, 6 or 8 Gy TBI. A total of 28 metabolites, including amino acids, organic acids and fatty acids, were selected as early-responding metabolites for different radiation dose groups. The differential effect on the metabolites was compared over three time points; 5, 24, and 48 h post-irradiation; with the most notable differences measured at 48 h. Several of the metabolites, including TCA cycle intermediate metabolites (fumaric acid, succinic acid and oxoglutaric acid), branched chain amino acids (l-leucine, l-isoleucine), citrulline, and hippuric acid were variably decreased at all three time points after radiation exposure. A small subset of metabolites, including oxalic acid, phosphoric acid, l-threonine, l-aspartic acid and saturated free fatty acids (myristic acid and palmitic acid), were elevated in the irradiated groups. Most of the measured, urine-based metabolites showed both dose- and time-dependence in response to radiation exposure. These observations illustrate how trends in metabolic dysregulation can be observed using a multi-criteria approach of mass spectrometry techniques, and their applicability to early triage following radiation exposure [169].

### 5.3. Differential Ion Mobility Spectrometry (DMS-MS)

DMS-MS separates metabolites based on attraction between the ion and the neutral drift gas molecule traveling through the ion filter in 5–10 msec. This ion-neutral clustering reduces the chemical background and the signal-to-noise ratio, thereby allowing the separation of ions with similar m/z, by using differences in chemical structure or charge states. These features make this technology fast, sensitive, and easy to use without the need to tune parameters [16]. While the value of LC-MS technology has been described above, there are still limits to this technology, in terms of its ability to rapidly screen large numbers of samples after a public health emergency. DMS-MS is capable of faster, easier to use screening diagnostics that are scalable, leading to a more feasible triage tool. Unlike the liquid phase separation technique used in chromatography, the separation method in DMS occurs in the gas phase during passage through the ion filter, reducing transit time. The technique is based on the ionic properties of the sample-structure, polarizability and cluster affinity. Thus, DMS provides orthogonal separation and works as a prefilter to MS. DMS-MS has been used as a high-throughput alternative to LC-MS for quantitative analysis in a variety of research fields, including biomarker discovery [129].

In a study using the planar DMS-MS method, seven metabolites were identified from NHP urine 7 days after gamma radiation exposures of 0, 2, 4, 6, 7, or 10 Gy. Prior to DMS, the stable-isotope-dilution (SID) analytical method (consisting of sample preparation by strong cation exchange-solid phase extraction) was used to concentrate the metabolites of interest. DMS was then employed for the selection of the target ion and to reduce background interference, followed by MS. The SID-SPE-DMS-MS methodology allowed for calibration curve preparation in under two hours for six simultaneously analyzed radiation biomarkers. The study used SID-SPE-DMS-MS, which has 7.5 to 30 times higher throughput than that of LC-MS. Samples were originally analyzed using LC-MS [126], and in this study, they were reanalyzed using DMS-MS. General agreement was noted in creatinine, xanthurenic acid, creatine, and hypoxanthine levels, but effects in xanthine were different. Although similarities were found, DMS-MS demonstrated a clear dose-dependent increase for creatine, trimethyl lysine, xanthine, and xanthurenic acid, especially in the transition from 6 Gy to 10 Gy. Such quantitative biomarker analysis can be used to determine radiation dose exposure and potentially estimate the consequence of kidney, liver and pancreas dysfunction [129].

The use of DMS-MS for biomarker detection has also been reported for resolution of test mixtures of isobaric compounds, separation of charge states, separation of isobaric biomarkers (citrate and isocitrate), and analysis of biofluids from mice exposed to radiation. Researchers here used a series of experiments to test DMS performance in interrogating selectivity, sensitivity, and the ease of use of DMS-MS for biomarker detection. DMS selectivity includes the ability to separate isobaric compounds, reduce chemical noise, and separate charge states, and has greater orthogonality than time-of-flight ion mobility spectrometry (IMS). For sensitivity, DMS involves short residence time (1–4 msecs), continuous operation without ion losses, and filtration of intact molecular ions. The ease of use of DMS is attributed to ability to operate in transparent mode (allowing ions to pass through the filter with minimum attenuation), constant intensity (DMS resolution is almost independent of operating parameters separation voltage (SV) and compensation voltage (CV)), and polarity independence (DMS filters ions of both polarities, no electrical changes are necessary). This performance evaluation suggests that DMS-MS is a promising candidate for rapid and powerful biomarker assessment, although additional testing and development are still necessary [17].

### 5.4. Capillary Electrophoresis–Mass Spectrometry (CE-MS)

CE-MS is yet another method that provides very high resolution for a wide range of polar and charged metabolites (70–1027 m/z), since compounds are separated by charge-to-size ratio [170]. This technique uses simple, fused-silica capillaries rather than expensive LC columns, to obtain fast and efficient analysis. In addition, sample preparation is straightforward and requires little to no consumption of organic solvents or other reagents [171]. A recent study used CE-MS to measure changes in the levels of blood metabolites in mice after exposure to radiation. C57BL/6 mice were exposed to 0, 1, or 3 Gy radiation, followed by whole blood collection on day 2 and 6 post-irradiation. Plasma was separated from the blood cells and a blood cell suspension was analyzed via CE–TOF-MS. Of the 306 metabolites detected, levels of 100 metabolites were significantly changed after irradiation. Aspartic acid, urea, and creatinine metabolism and neurotransmitter-related metabolites (aspartic acid, tyrosine, choline, homovanillic acid, and γ-aminobutyric acid) decreased after exposure. In addition, 2′-aminobutyric acid, 2′-deoxycytidine, and choline were also identified as potential biomarkers of ionizing radiation [172]. When compared to other studies, a recent NHP study also detected decreased levels of aspartic acid, glycine, and tyrosine, but other levels showed the opposite response [127]. Such findings highlight the importance of understanding the chosen MS technology, as well as the sample type (e.g., blood, plasma, urine, tears, etc.) being used for analysis.

While traditional detection methods are evolving even further to meet the demands of an emergency response, researchers are also tapping into unique sources, rethinking methods, and looking for clues in small molecules. MicroRNAs (miRNAs) and metabolites are two small molecules that are being explored, since both are readily available in the blood serum. In addition, the expression of both miRNAs and metabolites are dramatically altered in the early phase of radiation exposure [173]. In fact, in a recent study published in *Nature,* microRNA and metabolite signatures were analyzed in tandem, and miRNAs appear to regulate metabolites [173]. Using PCA analysis, miRNAs clustered and showed clear separation between three phases post-irradiation; and biological networks were associated with all phases—cardiomyopathy, cell proliferation, inflammation, tumorigenesis and carcinogenesis. Metabolite profiling was achieved using quadrupole time-of-flight (Q-TOF) mass spectrometry (MS) with electrospray ionization; this revealed that expressed metabolites also clustered into three phases post-irradiation. Given that miRNA regulate and influence the epigenetic landscape, it is no surprise that miRNAs can perturb metabolomic expression and/or homeostasis [174]. A time lag was noted between the miRNA expression and the metabolite profile, suggesting that miRNAs react to the radiation insult, and exert change on the metabolic profile. In fact, an miRNA-metabolite integrative network revealed a marked early inflammatory response and subsequent networks in later phases that related to comorbidities, such as cardiovascular dysfunction, bone marrow aplasia and cancer (tumorigenesis and oncogenesis) [173]. These exciting data, combined with giant leaps in technological advancement and agile bioinformatics, highlight the possibility for a multiparametric omics approach to radiation biodosimetry.

## 6. Analytical Methodology

Like other “omics” analytical approaches, metabolomics is a fairly new discipline. Nearly 95% of the approximately 38,000 scholarly articles on the topic over the last 20 years have been published since 2009. This explosion of interest in the field has been driven by its potential to shine a light on the biological end state metabolite profiles representing a number of key biological activities (gene expression, transcription, enzyme activities, cell signaling, etc.) [175,176,177]. Swift and dynamic changes in biological processes and responses to stimuli are reflected by variations in the output and end products of these processes, in the form of changes in the metabolite profile, or metabolome [175]. Recent technological advances in NMR, LC, and MS significantly improved the quantity of data obtained from each biological sample. Consequently, fast and accurate statistical and bioinformatic tools are required to process the complexity and volume of the metabolomic data generated. Both NMR and LC techniques use univariate and multivariate statistical analysis to identify major trends in the metabolomic changes, and to arrive at a signature. Several bioinformatic techniques are employed, including principal component analysis (PCA), random forest machine learning algorithm, GEDI (gene expression dynamics inspector) algorithms that use self-organizing maps, and orthogonal partial least squares [8,14,16] and reviewed by Coy et al. [16] A more recent review by Lamichanae et al. describes in detail the data preprocessing approaches based on the platforms used for data gathering, the advantages and weak points in targeted and untargeted metabolomic assays, and univariate and multivariate statistical tools to extract the meaningful interpretation of data [178]. Varghese et al. opined that for reliable identification of metabolites truly associated with radiation, these analytical techniques must be used in combination with a deep knowledge of the biology and biochemistry of metabolic pathways; this results in the validation of a small number of metabolic markers for radiation exposure [179].

## 7. Databases

Biological sample analysis by various instruments can provide values, but these values must first be identified and quantified to determine a radiation effect on the whole organism or organ systems. Complex biological sample analysis will lead to a complex series of peaks, and these peaks need to be differentiated, identified, and quantified. Computer algorithms can be used for peak identification, along with searches of databases of known libraries [178]. Because they are small molecules, metabolites tend to be similar between animal species used in radiation research and humans. For this reason, library searches are often performed using major databases such as the Human Metabolome Database (HMDB) [180,181,182], the Kyoto Encyclopedia of Genes and Genomes (KEGG [183]), Lipid Maps Database [184], and METLIN [185]. A more thorough analysis can be performed using tools such as MetaboAnalyst (at this writing, v4.0) [186]. Meta-analysis of data sets from different studies allows for broadening the results, adding to the robustness of the data sets. Data sets can be complex, spanning a variety of different metabolites in samples collected over time, and combining the results of different studies only adds to the complexity. These large data sets, and the combining with genomics to understand systems biology, lend themselves to a Big Data Analytics approach. Using advanced computer power and algorithms, investigators can discern patterns, and by studying various data sets, can determine whether these patterns hold true over multiple data sets [187].

Most of the basic biomedical research performed in the United States is funded by the NIH. Since this research is supported by public funding, the NIH has increasingly committed itself to the principle of resource and data sharing. In 2003, NIH’s policy [188] stated that it “expects and supports the timely release and sharing of final research data from NIH-supported studies for use by other researchers.” Grant applications submitted since then have been required to include a data sharing plan. Beyond the concept that publicly funded studies should be made freely available, data sharing allows for different analyses and interpretations to add to the original use of the data, and allows for the important step of research validation, to bolster conclusions reached by the original investigator. While requiring data sharing, the NIH does recognize that data would be shared only after allowing for exclusive use by the investigator.

The NIH Common Fund has established a metabolomics consortium to advance metabolomics research and data sharing [189]. At this writing, one part of the consortium is the Metabolomics Workbench, hosted by the University of California, San Diego, which contains a data repository, as well as links to metabolomics libraries [190]. As another example of a way that data sets can be made available to the research community, NIAID has developed a Web portal called ImmPort. Although the focus of the system is immunological, this collection can also be adapted to other data sets from NIAID-funded studies. The RNCP at NIAID plans to make available a wealth of data—not just metabolomics, but proteomics and histological data—from animal model studies it has sponsored, so that connections and patterns may be analyzed, particularly for the multi-organ injury that is characteristic of radiation damage. In this way, it could be possible to integrate disparate data sets to understand the systems biology of the model, including the characterization of radiation-induced injury over time, and even across species.

## 8. Metabolomics and Radiation Countermeasures—The Regulatory Landscape

To reiterate, there are no radiation biodosimetry approaches or devices cleared by the FDA for use in the event of a mass casualty radiological incident at the writing of this manuscript. The technology and data to translate radiation signatures from small mammals to NHPs to humans are indeed attractive; however, key lessons can be learned from those that have successfully navigated the regulatory pathway utilizing metabolomics for other disease conditions, and can inform radiation biodosimetry development. The science of metabolomics has proven useful in the identification of biochemical pathways that help to understand physiological mechanisms of health and disease. As these pathways are being elucidated, researchers are using metabolomics in various aspects of drug development. For example, metabolomic biomarkers can be used to evaluate disease progression in an animal model for comparison to human pathology, metabolomics can play a role in elucidating a drug’s mechanism of action, and they can be used as toxicity biomarkers to evaluate drug safety [191]. Still, the science is more academic in practice, since drug companies are historically reticent to invest in the development of novel biomarkers that have no guarantee of furthering their pipelines. Similarly, regulatory health authorities (RHAs), while maintaining a keen interest in the scientific advances driven by metabolomics, have had comparatively little commentary on the topic. Consequently, regulatory guidance on general “omics” data, let alone metabolomics, is limited and provided on a case-to-case basis [192]. However, RHAs have been quite clear that, prior to acceptance, all biomarkers must have evidence for their intended use, as well as being qualified and validated [193].

In 2004, the U.S. FDA released the initial Critical Path Challenges and Opportunities Report, followed in 2006 by FDA’s publishing of the Critical Path Opportunities Report [194,195]. Metabolomics was discussed as playing a major role in developing safety biomarkers related to cardiac, kidney, liver and vascular toxicities, as well as providing evaluation tools for developing disease models and efficacy. Recognizing the complex path to take an exploratory biomarker to qualification for clinical use, the FDA implemented a Pilot Process for biomarker qualification [196,197]. The process is focused on addressing uncertainties that limit the validity of biomarkers in preclinical and clinical analyses. Uncertainties exist, such as the relationship between safety and efficacy benefit versus additional test burden, limitations on biomarker sensitivity and specificity and context, and the relevance to a specific end point [196].

A relatively new branch of metabolomics, pharmacometabolomics (PMb), along with the related fields of pharmacogenomics (PGx) and pharmacogenetics (PGt), describe the use of biomarkers to study the safety and efficacy of potential drug products [193,196,198]. Together, they are an important part of a systems biology approach to understanding disease pathophysiology and response to treatment [192,199]. In terms of drug development, the systems biology approach begins as early as the discovery phase, with the characterization of the target and markers for disease, treatment, and prognostics [191]. Moving into the preclinical phase, PMb, PGx, and PGt biomarkers provide insight into to the drug mechanism of action and markers of efficacy, and perhaps more importantly, in identifying safety and toxicity markers. In addition to the drug development spectrum, metabolomics promises an impact in the development of companion diagnostics [200], with the objective being to use biomarkers to match patient populations to effective therapeutics. Together, the biomarker data contribute to effective clinical trial design, and early implementation is encouraged by regulatory authorities [201].

How one plans to include metabolomics into their drug development program should begin in the early phases, with product characterization and analytical development. This is also the time to consider the quality and regulatory aspects that would impact product and, eventually, clinical trial design. The concept of quality by design (QbD) is described in the ICH Quality Documents, Q8 (R2) “Pharmaceutical Development”, Q9 “Quality Risk Management”, and Q10 “Pharmaceutical Quality System” [202,203,204,205]. In brief, QbD provides guidance on early planning, to include quality requirements such as early characterization of products and processes. For a novel biomarker, one is encouraged to review FDA’s Biomarker Qualification Program; details are available through FDA’s website (www.fda.gov). In accordance with QbD, qualification, demonstrating suitability for context of use, is a key step in determining critical quality attributes (CQAs). Once the CQAs are defined, methods for controlling them need to be developed, qualified, and validated. Current agency thought is provided in the Guidance for Industry for Bioanalytical Method Validation [205] as well as the ICH Guideline Q2(R1) [206].

A crucial guidance specific to the field of radiation biodosimetry, entitled ‘Radiation Biodosimetry Medical Countermeasure Devices–Guidance for Industry and Food and Drug Administration Staff’, was published in 2016 [42], to guide the frontrunners in advancing their biodosimetry tests for eventual FDA clearance. This document addresses aspects of device development; from sample collection and preparation, to assay performance and validation, to device specifications, and analysis. Far more importantly, to the radiation research community that conducts most studies in animal models, the guidance provides an Animal Model Consideration section that defines appropriate animal models, confounders in animal studies, and bridging animal data to human data (Figure 1).

Key resources that are often overlooked include the regulations and associated guidance for computer systems, electronic records, and data integrity (for example, 21 CFR Part 11 and EU’s Annex 11). The design and validation status of the electronic systems used for collecting, handling and storing data will greatly impact the reliability, integrity, and robustness of metabolomics data, as well as that for product and process development. Failure to document the regulatory compliance of electronic records and computer systems may raise flags for RHA reviewers. Finally, it is wise to establish communications with the RHA early in the development process. These communications will provide valuable feedback and help with the development of an effective regulatory strategy for using metabolomic biomarkers.

## 9. Conclusions

The emerging field of radiation metabolomics provides a fascinating alternative to classical radiation biodosimetry approaches. Although promising, the literature still demonstrates a need for standardized protocols, validation of targeted biomarkers and metabolomic pathways for successful translation, and ultimately, to FDA clearance. We have summarized the role of sample preparation and collection, and several common preexisting conditions that can influence the metabolomic signatures in early discovery stages and confound advancement of targeted approaches. The choice of instruments and technologies, to data preprocessing steps, and tools for data analysis, are key in allowing discrimination between the exposed and the non-exposed cohort, using non-invasive biosamples. As described in the NIH policy on rigor and reproducibility [207], it is strongly encouraged that the researchers employ robust approaches to optimize data generation, and consistent data analysis. Cross validation of these findings will preclude false discovery and allow accurate biological interpretation from a metabolomic signature. Keeping abreast with cutting edge computational tools and knowledge-based adjustments will allow researchers to deconvolute complex biological radiation responses to usable, triage and clinical tools.

## Figures and Tables

**Figure 1 metabolites-10-00328-f001:**
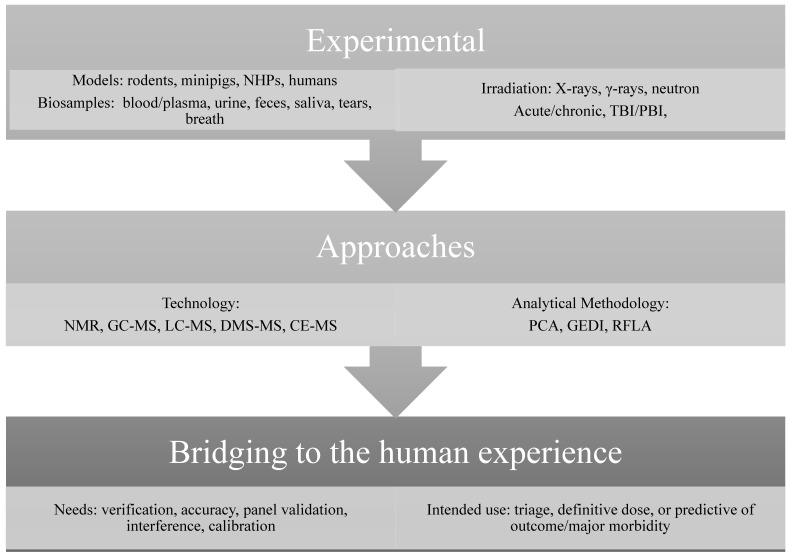
An overview of the radiation metabolomics biodosimetry pathway. Abbreviations: TBI-total body irradiation, PBI-partialbody irradiation, NMR-nuclear magnetic resonance; LC/MS, liquid chromatography/mass spectrometry; GC/MS, gas chromatography/mass spectrometry, DMS-differential ion mobility spectrometry, CE-capillary electrophoresis, PCA-principal component analysis, RFML-random forest machine learning algorithm, GEDI-gene expression dynamics inspector.

**Table 1 metabolites-10-00328-t001:** Attributes for biodosimetry approaches being developed by NIAID/RNCP.

Specifications	Point of Care Device	Definitive Dose Device	Predictive Biodosimetry Device
CONOPS	Triage	Dose for medical management	Dose/injury for medical management
Result	Qualitative/Semi-quantitative	Quantitative	Qualitative/Quantitative
Dose range	≤2 Gy	0.5–10 Gy	0.5–10 Gy TBI; >6 Gy PBI
Ease of use	Simple, minimal technical requirement	High degree of automation, laboratory based	Laboratory assay
Number of tests	1 M/week	40,000/week	10,000/week
Time to results	Rapid sample to answer in 15–30 min	24 h	≥24 h, may require longitudinal sampling

CONOPS = Concept of Operations, TBI = Total Body Irradiation; PBI = Partial Body Irradiation.

**Table 2 metabolites-10-00328-t002:** Trends in metabolites following Total Body Irradiation Exposure.

Metabolite	Species	Radiation	Trend	Reference
Citric Acid		X-, γ-rays	↓	[14,116,124]
Citrulline		X-, γ-rays	↓	[114]
Creatine		X-rays	↑	[124]
Taurine		X-, γ-rays, neutron	↑	[14,36,124,128]
Carnitine	Mouse	γ-rays	↓	[128]
Xanthine		γ-rays	↑	[14]
Creatinine		γ-rays	↓	[45]
Hypoxanthine		γ-rays	↑	[145]
Uric Acid		X-, γ-rays, neutrons	↑	[36,128,155]
Threonine		X-rays	↑	[114]
Glycoxylate		X-, γ-rays	↑	[131]
Tyramine sulphate		γ-rays	↑	[18]
Citric Acid		γ-rays	↓	[154]
Citrulline		γ-rays	↓	[26]
Creatine		γ-rays	↑	[18]
Taurine	NHP	γ-rays	↑	[25]
Carnitine		γ-rays	↑	[125,126]
Xanthine		γ-rays	↑	[18,129]
Creatinine		γ-rays	↓	[18,129]
Hypoxanthine		γ-rays	↑	[18,21]
Uric Acid		γ-rays	↑	[18]
Threonine		γ-rays	↑	[126]
Xanthine		X-rays	↑	[22]
Uric Acid	Human	X-rays	↑	[22]

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
