# Peer review of "Metabolomics in Radiation Biodosimetry: Current Approaches and Advances"

_metabolites, 2020, doi:10.3390/metabo10080328_

Round 1

Reviewer 1 Report

The review by Satyamitra et al titled “Radiation Metabolomics: Current Approaches and Advances” provides a comprehensive summary update about metabolomics based applications for biodosimetry. The review adeptly reviews the current status of metabolomics technologies, potential biomarkers reported by different researchers and potential confoundings for acute radiation syndrome. Since NIH has funded metabolomics based biodosimetry grants, it would be good to touch upon the following points:

  1. What has prevented the development of FDA biodosimetry metabolomic panels despite an array of published studies?
  2. What is needed to achieve this in compliance with biomarker qualification guidelines stipulated by FDA?
  3. What technologies are most suited for implementation of these panels for field deployment?
  4. Since most of the studies are based on animal models, how feasible is it to translate these biomarkers for human use?
  5. It would be nice to have a table in the review that summarizes the most promising biomarkers, especially those that show overlap between different doses and times of exposure and across species.
  6. The review does not touch upon radiation late effects and the utility of metabolomics approach for development of predictive markers of organ injury in asymptomatic individuals.
  7. Minor point: the authors have heavily cited work of a few labs. rather than a broad based contribution of different labs to this field

Author Response

The review by Satyamitra et al titled “Radiation Metabolomics: Current Approaches and Advances” provides a comprehensive summary update about metabolomics based applications for biodosimetry. The review adeptly reviews the current status of metabolomics technologies, potential biomarkers reported by different researchers and potential confoundings for acute radiation syndrome. Since NIH has funded metabolomics based biodosimetry grants, it would be good to touch upon the following points:

1. What has prevented the development of FDA biodosimetry metabolomic panels despite an array of published studies?

From a review of literature, there is a lack of data that translates from mice/rats to monkeys to men consistently over different radiation doses and time-points. As of the writing of this article, the authors are not aware of any metabolomic-focused technology that has been proposed as a biodosimetry TEST to the FDA.

2. What is needed to achieve this in compliance with biomarker qualification guidelines stipulated by FDA?

For biodosimetry clearance, the CDRH/FDA released a guidance document that highlights the requirements of validation and verification studies required to advance the biodosimetry test. We have included this guidance as part of the references under ‘Regulatory’ section.

3. What technologies are most suited for implementation of these panels for field deployment?

According to the US DHHS Biodosimetry and Radiological/Nuclear Medical Countermeasures Program, “A Point-of-Care test for a quick triage of an exposed population based on a biological dose above or below 2-Gy to determine risk of developing ARS. This test is designed to be administered by a person with little or no medical training in a field hospital, tent or triage station. The point- of-care device is expected to be able to screen up to a million individuals for determining a cohort at-risk for developing ARS. Ideally, results would be available in less than 15 min after the sample is collected with little or minimal sample processing. The intent of this assay is to separate the individuals with radiation-related medical needs from the ‘concerned citizens’ who may not need specific treatment.” Based on the specifications that need a field deployable POC to be easily deployable, the metabolomic assay does not appear suitable for field deployment, but could be more amenable to assessing definitive dose.

Table 1 described the requirements for triage, treatment and prediction and is now included in the manuscript.

4. Since most of the studies are based on animal models, how feasible is it to translate these biomarkers for human use?

It is advised that the investigators approach the FDA often and early in their development pathway. Once they identify the panels in lower mammals and translate this to higher mammals, they will be required to validate and verify their panel in irradiated clinical samples to demonstrate translation, and ultimately to obtain FDA approval. Language to this effect is included on line 118-123. The FDA guidance on radiation biodosimetry device is also included as reference.

5. It would be nice to have a table in the review that summarizes the most promising biomarkers, especially those that show overlap between different doses and times of exposure and across species.

The authors concur with the reviewer on this point.

We have included a table with relevant references and trends in metabolites in Table 2.

We have also included a recent publication by Vincente et al (2020) that contains this exact information in great detail and is referenced  lines 315-317.

6. The review does not touch upon radiation late effects and the utility of metabolomics approach for development of predictive markers of organ injury in asymptomatic individuals.

Indeed this is a fact. We have included the significance of TBI vs PBI Ln 113-117. Upon review, we have found just one paper (Liakis et al 2019) identify salivary metabolites in irradiated NHPs upto 60 d post-exposure. However, the PIs have stated this observation but did not ascribe a ‘predictive’ function to the panel other than indicating that there was significant reduction in certain amino acids observed months following exposure. We have included this statement on line 325. Boerma and co also investigated cardiac injury biomarkers in irradiated mice, which is also referenced on Ln 328-330.

7. Minor point: the authors have heavily cited work of a few labs. rather than a broad based contribution of different labs to this field

Although we tried to be as comprehensive as possible, we may have missed some publications/groups. We have revisited this issue and included some references that we missed such as Menon et al 2016a, Vincente et al 2020.

Reviewer 2 Report

Metabolites-870605

Title:  Radiation Metabolomics: Current Approaches and Advances

Authors: Satyamitra MM et al.

General Comments. The authors have written a review of the field of “radiation metabolomics” addressing a broad scope of relevant sub-topics including: a) influence of intrinsic (i.e., sex, age, etc.), disease, and extrinsic (i.e., smoking, obesity, exercise, etc.) factors; b) procedural testing; c) bioanalytical, sampling and detection technologies, d) databases and analysis methodologies, and the e) regulatory landscape of metabolomics and radiation countermeasures.

Overall the contents of the review reported here are of interest to the radiobiology community. Major comments intended to improve the review include:

  • Add critical needed item (partial-body assessment) to the toolkit for first-responders and health-care providers.
  • Need to briefly discuss other relevant reviews on “radiation metabolomics” for radiation exposure and injury assessment.
  • Need to briefly discuss relevant manuscripts addressing use of “volatile breath” biomarkers for radiation dose and injury assessment.

Specific Comments.

Page 2, 1st paragraph. The list of needed items in the toolkit is missing an important one; ability to determine, in individuals exposed to life-threatening doses, if there is complete bone marrow aplasia or up to 5% bone marrow sparing (Dainiak N et al. Concept of operations for a US dosimetry and biodosimetry network, Radiat Prot Dosim 186(1): 130-138, 2019.

Page 2, 3rd paragraph, lines 60-61. The field of radiation metabolomics has a broad scope and only one component focus on the assessment of the level of radiation exposure. For example, see review by Gramatyka and Sokol (Radiation metabolomics in the quest of cardiotoxicity biomarkers: the review, IJRB 96 (3): 1-3, 2019) addressing cardiotoxicity biomarkers.

Page 2, 3rd paragraph, lines 65-67. Briefly address other relevant review articles on the topic of “radiation metabolomics”. See Vicente E et al. (A systematic review of metabolomic and lipidomic candidates for biomarkers of radiation injury, Metabolites 10(6): 259, 2020), Roh C (Metabolomics in radiation-induced biological dosimetry: a mini-review and a polyamine study, Biomolecules 8, 34, 2018), Menon SS et al. (Radiation metabolomics: current status and future directions, Front Oncol 6: 20, 2016).

Page 6, 1st paragraph, line 248-250. Clarify the purpose for “some metabolite signatures” (i.e., radiation injury, dose, etc.).

Page 7, Section 3.5. Briefly address relevant manuscripts on breath biomarkers; see Phillips M et al. (Detection of volatile biomarkers of therapeutic radiation in breath. Journal of Breath Research 7(3); 1-3, 2013; Breath biomarkers of whole-body gamma irradiation in the Gottingen minipig. Health Physics 108(5): 538-46, 2015).

Page 9, 1st paragraph. Derivatization to add trimethylsilyl functional groups are commonly done in GC analysis to convert non-volatile compounds into volatile derivatives suitable for GC analysis; add to clarify.

Editorial Comments.

Page 6, line 244. Dose reconstruction, injury diagnosis, and prognosis.

Author Response

General Comments. The authors have written a review of the field of “radiation metabolomics” addressing a broad scope of relevant sub-topics including: a) influence of intrinsic (i.e., sex, age, etc.), disease, and extrinsic (i.e., smoking, obesity, exercise, etc.) factors; b) procedural testing; c) bioanalytical, sampling and detection technologies, d) databases and analysis methodologies, and the e) regulatory landscape of metabolomics and radiation countermeasures.

Overall the contents of the review reported here are of interest to the radiobiology community. Major comments intended to improve the review include:

  • Add critical needed item (partial-body assessment) to the toolkit for first-responders and health-care providers.

‘Partial body’ term included on line 48, pg 2 in the revised draft, along with Daniak et al, 2019 reference.

  • Need to briefly discuss other relevant reviews on “radiation metabolomics” for radiation exposure and injury assessment.

Reviews relevant to radiation biodosimetry metabolomics are included in the revision. The authors thank the reviewer for pointing out some important articles of significance.

  • Need to briefly discuss relevant manuscripts addressing use of “volatile breath” biomarkers for radiation dose and injury assessment.

Specific Comments.

Page 2, 1st paragraph. The list of needed items in the toolkit is missing an important one; ability to determine, in individuals exposed to life-threatening doses, if there is complete bone marrow aplasia or up to 5% bone marrow sparing (Dainiak N et al. Concept of operations for a US dosimetry and biodosimetry network, Radiat Prot Dosim 186(1): 130-138, 2019.

‘Partial body’ term included on line 48, pg 2 in the revised draft, along with Daniak et al, 2019 reference.

Page 2, 3rd paragraph, lines 60-61. The field of radiation metabolomics has a broad scope and only one component focus on the assessment of the level of radiation exposure. For example, see review by Gramatyka and Sokol (Radiation metabolomics in the quest of cardiotoxicity biomarkers: the review, IJRB 96 (3): 1-3, 2019) addressing cardiotoxicity biomarkers.

The authors agree with the reviewer that the field of radiation metabolomics has a wide scope. We have focused entirely on these insular facets since this is most relevant to our mission. We have changed the title to “Metabolomics in Radiation Biodosimetry: Current Approaches and Advances’ to emphasize our focus.

Publications by Gramatyka and Co focus on metabolites in irradiated cardiomyocytes and cardiac tissue, whereas we are focused on easily accessible biosamples such as fingerstick blood/plasma/serum, saliva, urine, sweat, rather than biopsy samples. This is described in lines 309-312. Rather we included a reference by Unger et al, 2020, wherein plasma is used in predicting radiation-induced cardiotoxicity (ln 328-330).

Page 2, 3rd paragraph, lines 65-67. Briefly address other relevant review articles on the topic of “radiation metabolomics”. See Vicente E et al. (A systematic review of metabolomic and lipidomic candidates for biomarkers of radiation injury, Metabolites 10(6): 259, 2020), Roh C (Metabolomics in radiation-induced biological dosimetry: a mini-review and a polyamine study, Biomolecules 8, 34, 2018), Menon SS et al. (Radiation metabolomics: current status and future directions, Front Oncol 6: 20, 2016).

The authors thank the reviewer for raising this point. We have included these references on lines 69-70.

Page 6, 1st paragraph, line 248-250. Clarify the purpose for “some metabolite signatures” (i.e., radiation injury, dose, etc.).

We have included the phrase ‘over a range of radiation doses’ on line 326 to clarify.

Page 7, Section 3.5. Briefly address relevant manuscripts on breath biomarkers; see Phillips M et al. (Detection of volatile biomarkers of therapeutic radiation in breath. Journal of Breath Research 7(3); 1-3, 2013; Breath biomarkers of whole-body gamma irradiation in the Gottingen minipig. Health Physics 108(5): 538-46, 2015).

These references are included in the revised version from lines 407-413.

Editorial Comments.

Page 6, line 244. Dose reconstruction, injury diagnosis, and prognosis.

The changes have been incorporated on line 320 in the revised version.

Reviewer 3 Report

Authors of this review attempt to provide an overview highlighting the current approaches and advances in radiation metabolomics.

Unfortunately, I am not convinced that the authors have given serious thoughts in reviewing the literature to provide a summary that will significantly benefit the readers of Metabolites. As mentioned in the Introduction, "the authors have investigated the literature to ascertain best practices in the field of metabolomics, in an effort to provide a reference document for use by researchers to further improve and accelerate translation of this key technology" - I agree with this sentence but my concern is that the radiation component is lacking. In the major part of the paper, the word "radiation" could be changed to any other stimulus and this would not change the perception of the text (more radiation context could be found only in part 4 of the review).

Also, I would expect figures and tables to help guide the reader through the text. Key points summarizing current findings and highlighting future directions/acute needs would be also useful.

As mentioned above, in my opinion, this review is too general and provides only very broad perspective on the whole field of metabolomics with no particular focus on radiation-related issues. Adding this new review in this form to the vast body of the already available literature can be moderate.

Author Response

Authors of this review attempt to provide an overview highlighting the current approaches and advances in radiation metabolomics.

Unfortunately, I am not convinced that the authors have given serious thoughts in reviewing the literature to provide a summary that will significantly benefit the readers of Metabolites. As mentioned in the Introduction, "the authors have investigated the literature to ascertain best practices in the field of metabolomics, in an effort to provide a reference document for use by researchers to further improve and accelerate translation of this key technology" - I agree with this sentence but my concern is that the radiation component is lacking. In the major part of the paper, the word "radiation" could be changed to any other stimulus and this would not change the perception of the text (more radiation context could be found only in part 4 of the review).

The authors thank the reviewer for their input. We agree that we delved heavily into the broad field of metabolomics, several without radiation components. As proposed, we intended to peruse these publications to fill in the blank in the rather preliminary radiation biodosimetry focused metabolomics field. Upon reading your comment, we have returned to the text to fill in more radiation focused references, as applicable. The second section is now changed to variabilities arising from radiation source and animal models selected. Further, we have included recent references from Menon et al, Roh, and Vicente on page 2. We have also included more references under breath metabolomics on page 7 in the revised document. We have revisited each section and included relevant radiation references to them:

Age-Ln 153-154; smoking 192-193; obesity-ln 202-208; exercise-222-224; stress-235-237; nutrition-267-275; circadian rhythm-ln 284-286; 303-306; Breath biomarkers-408-413.

Also, I would expect figures and tables to help guide the reader through the text. Key points summarizing current findings and highlighting future directions/acute needs would be also useful.

Thank you for this excellent suggestion! We have included a figure that encompasses the concept of sample to test flow-chart (FIG 1). We have also included 2 tables to highlight the RNCP mission focus for biodosimetry and to summarize the major metabolites identified across species. We hope that this will prove to be a cogent addition to the manuscript.

As mentioned above, in my opinion, this review is too general and provides only very broad perspective on the whole field of metabolomics with no particular focus on radiation-related issues. Adding this new review in this form to the vast body of the already available literature can be moderate.

We have incorporated more radiation centric details, as well as our mission focus to the revision. Also included is the Biodosimetry Device guide, a key document describing for FDA clearance process for radiation biodosimetry regulatory processes. We have endeavored to meet your requirements by emphasizing our mission area and highlighting the radiation relevant references for each section.

Round 2

Reviewer 1 Report

Th authors have addressed all critiques.